# Theoretical Study on Performing Movement-Related MEG with $^{83}$Kr-Based Atomic Comagnetometer

Yao Chen *, Ruyang Guo, Jiyang Wang , Mingzhi Yu, Man Zhao and Libo Zhao *

School of Instrument Science and Technology, Xi'an Jiaotong University, Xi'an 710049, China;
3123301344@stu.xjtu.edu.cn (R.G.); jiyangwang@stu.xjtu.edu.cn (J.W.); yumingzhi@stu.xjtu.edu.cn (M.Y.);
zm1994@stu.xjtu.edu.cn (M.Z.)
* Correspondence: yaochen@xjtu.edu.cn (Y.C.); libozhao@xjtu.edu.cn (L.Z.)

**Abstract:** A K–Rb–$^{83}$Kr-based atomic comagnetometer for performing movement-related Magnetoencephalography (MEG) is theoretically studied in this paper. Parameters such as the spin-exchange rates, the spin-dephasing rates and the polarization of the nuclear spins are studied to configure the comagnetometer. The results show that the nuclear spin can generate a magnetic field of around 700 nT, at which the nuclear spin can compensate for a wide range of magnetic fields. In this paper, we also show the fabrication process for hybrid optical-pumping vapor cells, whereby alkali metals are mixed in a glove box that is then connected to the alkali vapor-cell fabrication system.

**Keywords:** atomic comagnetometer; spin-exchange optical pumping; optically pumped magnetometer; MEG; atomic spin gyroscope





## 1. Introduction

Magnetoencephalography (MEG) recordings have found various applications, including source localization of epilepsy-related seizures to aid in planning epilepsy surgery [1,2]; the study of the brain's response to specific external stimuli, which helps map the motor system [3]; and sensory areas [4], such as language, vision, etc. MEG signal recordings can also be used for the diagnosis of early-stage Alzheimer's disease [5] and the study of cerebral networks in tremor syndromes [6]. The typically used traditional MEG measurement equipment includes SQUIDs (super-conducting quantum interference devices), in which a very large helmet with liquid helium is used, which limits the application of magnetometers to the motor system, therefore, it is not feasible to perform MEG while a person freely moves their head [7,8].

With the development of the optically pumped magnetometer (OPM), which can work at room temperature and reach sensitivity equal to that of SQUID magnetometers [9,10], future MEG equipment might be wearable [11–13]. A wearable OPM system and EEG (electroencephalography) have been combined for the study of brain function [14]. This could reinforce more applications of MEG equipment, such as motor system-related space navigation of the brain and MEG for visual–motor integration.

Even though OPMs are wearable and movable, an important technique-related problem should be considered. Different from EEG, in which there is no large background electric field in the environment, MEG recording could be affected by the geomagnetic field of the earth. We know that the brain magnetic field is quite weak compared with the geomagnetic field. Thus, magnetic-field shield rooms (MSRs) are typically used for suppression of the environmental magnetic field. Moreover, the penetration of the magnetic field into MSRs can still affect MEG recording. In order to perform movement-related MEG, a large bi-planar coil was used for movement-related background-magnetic-field compensation [15]. During MEG recording, a person can only move in a small area, or only very small movements can be performed, such as drinking water [8]. In order to enable ambulatory movements in wearable MEG, matrix coil active magnetic shielding was used [16]. Some other methods based on magnetic-field

compensation have also been studied [17–19]. The magnetic-field differential method can also be used for background-magnetic-field suppression [20].

Even though many methods have been developed for movement-related MEG measurement, most of these methods are based on magnetic-field compensation in the space of the movement. Large compensation coils are required, which can make the system very complicated. Moreover, this method cannot compensate for the magnetic field of each sensor head, and some of the sensors could be over- or under-compensated.

As we understand, the crucial aspect of performing movement-related MEG lies in compensating for fluctuations in the magnetic field. The method based on hyperpolarized nuclear spins can also be used [7]. The nuclear spin method could compensate for the magnetic field in situ as well as automatically compensate for the fluctuations in the magnetic field in each of the OPMs. Complicated compensation coils are not needed, resulting in a significant reduction in the system's volume.

Hyperpolarized nuclear spins include those of $^{3}$He,$^{21}$Ne,$^{129}$Xe and $^{131}$Xe. The first demonstration of MEG measurement during movement was performed with $^{21}$Ne [7]. There is a small spin-exchange rate between $^{21}$Ne and alkali atoms such as K or Rb and the $^{21}$Ne nuclear spin features a very large quadrupolar relaxation rate [21]. In addition, $^{21}$Ne nuclear spins could not be easily polarized and hybrid pumping technique is required [22]. Moreover, the vapor cell temperature could reach 473 K for the polarization. The typically used boro-silicate glass could not withstand such high temperatures due to glass corrosion. The $^{21}$Ne nuclear spins are widely used for rotation sensing or the studying of fundamental physics [23–27]. However, due to the very low natural abundance of isotope-enriched $^{21}$Ne, it is very expensive to buy this gas. The $^{3}$He nuclear spins feature a very small spin-exchange optical-pumping rate with alkali metals; thus, it is very hard to polarize the nuclear spins of $^{3}$He. A hybrid pumping technique may be used. However, there is no report about the hybrid pumping of $^{3}$He for applications with atomic comagnetometers. Moreover, $^{3}$He nuclear spins are quite sensitive to magnetic-field gradients and can be easily depolarized in a holding magnetic field. Even the stem of a vapor cell could affect the spin polarization of nuclear spins [28–30]. It is worth noting that $^{129}$Xe and $^{131}$Xe can also be used. However, due to the large collision relaxation at the container walls, it is hard to polarize the nuclear spins of these elements [31,32]. For $^{83}$Kr nuclear spins whose atom mass is much smaller than that of Xe atoms, we can estimate smaller wall relaxation as well as weaker interaction strength for these alkali atoms. Moreover, $^{83}$Kr is easier to polarize than that of the $^{3}$He and $^{21}$Ne. In addition, $^{83}$Kr is a good candidate for movement-related MEG measurement. In this paper, we will focus on the study of $^{83}$Kr nuclear spins for background-magnetic-field fluctuation compensation in movement-related MEG measurement. We will first study the configuration of the $^{83}$Kr-based comagnetometer. The relaxation of $^{83}$Kr should be studied first since the relaxation could determine the polarization of the nuclear spins. Then the polarization of the nuclear spins is calculated. In order to acquire the sensitivity of the nuclear spin detection, we studied the Rb relaxation. The sensitivity of the comagnetometer will be calculated in the "Configuration of the Comagnetometer" section. In the Section 3, we studied the hybrid pumping cell fabrication. As we know, before we study the atomic comagnetometer, we need to first fabricate an alkali vapor cell. Hybrid pumping cells are needed. Thus, we have shown the method for hybrid pumping cell fabrication. The Sections 4 and 5 of the paper are the "Discussion" and "Conclusions".

## 2. Configuration of the Comagnetometer

### 2.1. Relaxation of $^{83}$Kr

The nuclear spin-based field compensation method for movement-related MEG is heavily related to nuclear spin polarization, which is closely related to the spin relaxation of nuclear spins. The $^{83}$Kr atom has a nuclear spin of $I = 9/2$; thus, it has a quadrupolar moment. We can say that quadrupolar relaxation is the main relaxation process of these nuclear spins. Moreover, container-wall-related quadrupolar relaxation of nuclear spins and quadrupolar relaxation in

$^{83}$Kr–$^{83}$Kr collisions are the main quadrupolar relaxation processes. There are several classical papers which illustrated the relaxation of $^{83}$Kr in detail; see [32–34].

Quadrupolar relaxation in $^{83}$Kr–$^{83}$Kr collisions is related to the number density of $^{83}$Kr atoms. The higher the number density of the nuclear spins, the faster the relaxation that is caused by quadrupolar interaction. It is reported that the relationship between quadrupolar relaxation time $T_1$ and number density of the nuclear spins $\rho$, in Amagat (1 Amagat is $2.69 \times 10^{19}$/cm$^3$), is $1/T_1 = \rho(2.13 \pm 0.05) \times 10^{-3}$s$^{-1}$ [34]. Under typical conditions of 50 Torr gas, the relaxation rate of gas-phase $^{83}$Kr collisions is $1.4 \times 10^{-4}$s$^{-1}$.

Quadrupolar relaxation at the container wall and spin-exchange relaxation of alkali atoms are the main relaxation processes and are around an order of magnitude higher than the $^{83}$Kr–$^{83}$Kr collision process. Since $^{83}$Kr has a quadrupolar moment, the nuclear spins can be relaxed by having them collide with the container wall of the vapor cell, where the impurities in the wall could produce electric-field gradients (EFGs). The spin-exchange optical-pumping process of alkali atoms can also cause the decay of nuclear spin polarization. The spin momentum can transfer between alkali atoms and $^{83}$Kr atoms. Note that the spin-exchange process can also be the pumping process, which can make the nuclear spins become polarized. We define the total relaxation rate of $^{83}$Kr nuclear spins [32] as

$$\left(\frac{1}{T_2}\right)_{m,n} = f_{m,n} < \omega_Q^2 > J(0) + \sigma_{ex}\bar{v}n_{Rb}, \tag{1}$$

where m and n represent the energy level, and in our study, we typically consider the m = 1/2 and n = −1/2 sublevels. For $^{83}$Kr nuclear spins, we calculate $f_{m,n} = 5/4$ using Equation (118) in reference [35].

$$< \omega_Q^2 > = \left(\frac{3e^2Q(1-\gamma_\infty)}{2\hbar I(2I-1)}\right)^2 < q^2 >, \tag{2}$$

which includes the mean square of the local quadrupole coupling constant ($< e^2qQ(1-\gamma_\infty)h^{-1} >^2$) as a measure of the mean interaction strength at the surface-wall site. In the reference, the coupling constant is measured to be 5.6 MHz for $^{83}$Kr atoms on typical Duran glass [32]. $I$ is the nuclear spin quantum number of $^{83}$Kr. $\hbar$ is Planck's constant. Note that $q$ is equal to $\sqrt{< q^2 >}$, which is defined to be the root mean square of the electric-field gradient. With these parameters, we can calculate $< \omega_Q^2 >$, which is approximately not temperature-dependent. $J(0)$ is defined to be the spectral density function of the normalized correlation function at zero frequency [36]. It is related to the stochastic sequence of wall collisions and thermally activated surface diffusion. Under our experimental conditions, it is believed that diffusion along the surface is strongly hindered and that quadruple relaxation is governed by the adsorption and desorption processes.

$$J(0) \approx \frac{\tau_s^2}{\tau_v} \propto exp(2E_A/k_BT) \tag{3}$$

In the above equation, $\tau_v$ equals $\bar{v}S/4V$, which is the average time it takes the nuclear spin to move to the wall. $\bar{v} = \sqrt{8k_BT/\pi M}$ is the average velocity of the nuclear spins. $M$ is the atom mass of the noble gas atom. $V$ is the volume of the vapor cell, and S is the overall surface of the vapor cell. $k_B$ is the Boltzmann constant. $E_A$ is the activation energy of desorption. We cannot directly obtain $\tau_s$, which is defined as the adsorption time of the nuclear spins on the surface wall. However, we can use the squared phase angle, $< \theta^2 >$, as in the reference [32], to acquire $\tau_s$. Using the relation $\tau_s^2 = (\tau_0 exp(2E_A/k_BT))^2 = < \theta^2 > / < \omega_Q^2 >$, we can calculate $\tau_s$ with the measured $< \theta^2 >$ at 373 K as $4.9 \times 10^{-8} rad^2$. Note that $\tau_v$ is related to $\bar{v}S/4V$. The cell shape and dimensions could determine the relaxation rate. In our simulation, we suppose that $4V/S$ is equal to 10 mm for approximation. This value is typical for vapor cells utilized in the hyper polarization experiments. It is also approximately equal to values in reference [32].

The total relaxation of $^{83}$Kr also includes the spin-exchange interaction with Rb atom spins, which will be considered in our study. The spin-exchange interaction cross-section ($\sigma_{ex}$), the velocity ($\bar{v}$) and the number density of Rb ($n_{Rb}$) determine the relaxation rate. Based on the spin-exchange term $\sigma_{ex}\bar{v}^2 = 1.9 \times 10^{-12} \mathrm{cm}^4\mathrm{s}^{-2}$ measured in the reference [32], we can obtain a numerical simulation of the total relaxation rate, as shown in Figure 1. The smallest relaxation rate happen at the temperature of 382 K and the relaxation rate is approximately 0.0021 s$^{-1}$. The relaxation time is around 500 s. This relaxation time is coincidental with the relaxation times acquired in these references [32,33]. Note that here we only consider the collision relaxation with Rb atoms and without considering the collision with the K atom spins. We have mentioned that K-Rb hybrid pumping would be utilized in the comagnetometer.

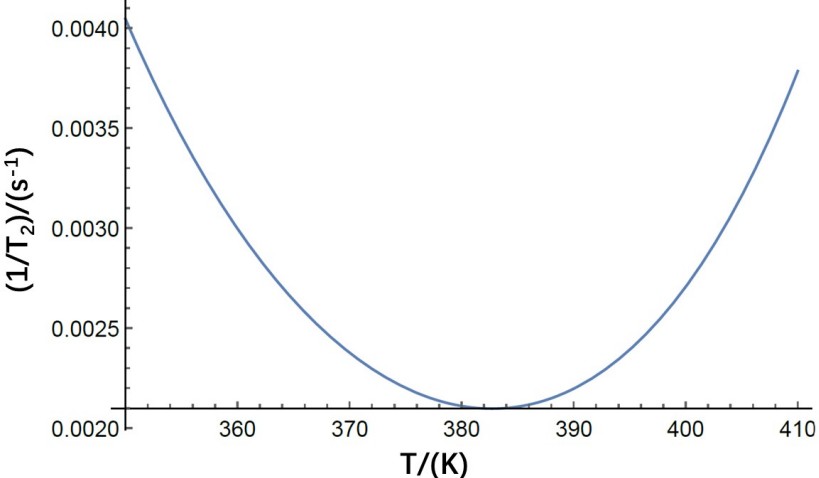

**Figure 1.** Numerical modeling of the dependent relaxation rate of nuclear spins on the temperature of the vapor cell. At low temperatures, $^{83}$Kr is adsorbed on the surface walls for a longer time, and fluctuation EFGs cause the relaxation of the nuclear spin. As the temperature increases, relaxation among collisions with Rb atoms is larger, because the number density of Rb is larger, and more alkali atoms depolarize the nuclear spins. The spin-exchange relaxation is dominant at high temperatures, since the atoms stay on the surface for a short time and the quadrupolar relaxation can be neglected.

In a hybrid optical pumping experiment, a mixture of the metals is typically used. According to the Raoult's law, the partial pressure of the saturated vapor of a metal above a liquid phase is determined by the composition of the mixture or the mole fraction ratio of a metal in the mixture. The partial pressure of a metal can vary from 0 to the value of the pure vapor pressure. As we know, the partial pressure of the metal could determine the density of the metal in the vapor cell. Thus, the density ratio of Rb to K could be determined. In the hybrid pumping experiment for the atomic comagnetoemter in which very large nuclear spin polarization is required, the density ratio of Rb to K is typically very large which means that most of the alkali atoms are Rb. The density ratio of Rb to K is more than 200 in these experiments [37].

In the hybrid pumping experiment, K is directly pumped by the laser and Rb is pumped by spin exchange interaction with K electron spins. As we have mentioned that the K density ratio is quite smaller than that of the Rb metal in our vapor cell design, the K medium could be designed to be optically thin (the optical depth is typically from 1 to 2, the temperature of the vapor cell could be changed to reach the required optical depth). Due to the large density ratio of Rb to K, the optical depth of Rb is larger than that of K. This means that Rb could not be directly pumped efficiently by a laser. The polarization gradient of Rb in the laser pass is quite large. However, due to the large collision cross section of the spin exchange interaction between K and Rb, the spin transfer rate from K to Rb is quite large. Rb could be uniformly polarized under hybrid pumping conditions. The

polarization gradient could greatly affect the spin polarization of the nuclear spin. A large polarization gradient could lead to the failure of an efficient nuclear spin hyper polarization. Note that even though we could allow the laser power to be as large as possible to reach a polarization of 1, we need to keep the alkali metal electron spin polarization at 0.5 to obtain the best sensitivity of the comagnetometer. We could see that in the hybrid pumping experiment, the Rb atomic spins are mainly used for spin exchange optical pumping of the nuclear spins. Due to the large density of Rb, some of the nuclear spins, such as $^3$He and $^{21}$Ne, which are very hard to polarize, could be efficiently polarized. It is reasonable to only consider the Rb-related collision relaxation in Equation (1).

### 2.2. Polarization of $^{83}Kr$

The polarization of the nuclear spin is very important, because it can determine the magnetic field experienced by the electron spins. With the relaxation and the spin-exchange optical-pumping rate of alkali atoms and nuclear spins, we can determine the polarization of the nuclear spins. In Equation (1), the term $\sigma_{ex}\bar{v}n_{Rb}$ can also be defined to be the spin-exchange optical-pumping rate of Rb and nuclear spins. The polarization of the nuclear spins can be defined as

$$P^n = P^e \frac{\sigma_{ex}\bar{v}n_{Rb}}{f_{m,n} < \omega_Q^2 > J(0) + \sigma_{ex}\bar{v}n_{Rb}}, \tag{4}$$

where $P^e$ is the polarization of the electron spin. We can see that the highest polarization of the nuclear spin can reach that of electron spins. It is known that the polarization of the electron spin can be 0.5, at which the magnetometer has the highest sensitivity. Thus, we let $P^e$ be 0.5. We performed a numerical simulation of the polarization of nuclear spins with the change in temperature. Figure 2a shows the results.

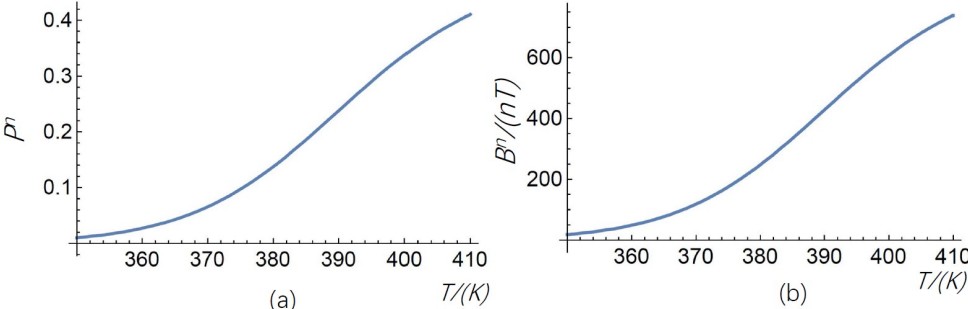

**Figure 2.** (**a**) Theoretical estimation of the relationship between temperature and nuclear spin polarization ($P^n$). As the temperature of the vapor-cell changes, we can estimate a rise in the spin-exchange optical pumping of the nuclear spins as well as a decrease in quadrupolar relaxation. Thus, the polarization of nuclear spins increases with the rise in temperature. (**b**) Theoretical estimation of the relationship between the temperature and the Fermi contact interaction enhanced magnetic field produced by the nuclear spin ($B^n$). As the magnetic field produced by the nuclear spin is directly related to nuclear spin polarization, we can estimate a similar change in $B^n$ with temperature.

As shown in Figure 1, the lowest relaxation falls at 382 K. We need to choose the appropriate temperature to minimize the relaxation rate as well as achieve very high nuclear spin polarization. Even though the lowest relaxation rate occurs at 382 K, however, as shown in Figure 2a, the polarization at 382 K is around 0.15—far less than the highest polarization, which is around 0.4. Thus, it is better for us to choose a working temperature at which the polarization of the nuclear spin is relatively higher as well as the boro-silicate glass could withstand such temperatures without corrosion of the glass. It is better for us to choose the temperature around 420 K.

In order to compensate for the fluctuation in the background magnetic field experienced by electron spins used in brain magnetic-field sensing, it is necessary to calculate

the magnetic field generated by the nuclear spins ($B^n$). We can estimate that the larger the nuclear spin magnetic field is, the higher the compensation ability of the nuclear spins is. The magnetic field produced by the nuclear spins is defined as

$$B^n = \frac{8}{3}\pi k_0 \mu_{Kr} n_{Kr} P^n,$$ (5)

where $k_0$ is the Fermi contact interaction enhancement factor, equal to 270 for the $Rb - ^{83}Kr$ pair [38,39]. $\mu_{Kr}$ is the magnetic moment of the $^{83}$Kr nuclear spin. $n_{Kr}$ is the number density of the nuclear spins. We supposed that the partial pressure of $^{83}$Kr gas inside the vapor cell was 50 Torr. We performed a simulation of the nuclear spin magnetic field with the change in temperature. Figure 2b shows the results.

The polarized nuclear spin could produce magnetic field $B^n$. This field could compensate the background disturbing field $B_y$ or $B_x$. As shown in Figure 3, the pump laser polarized the electron spins in the z direction and the electron spins could produce a similar magnetic field $B^e$ as that of the nuclear spin $B^n$. We can add a holding magnetic field which is constant in the z direction and the direction of this field is opposite to $B^n$ and $B^e$. The strength of this field is equal to $B^n + B^e$. Under this condition, the nuclear spin could automatically compensate the disturbing field in the x or y direction. If there is no disturbing field, both $B^n$ and $B^e$ will settle in the z direction. Due to the Fermi contact interaction, the nuclear spins could experience $B^e$. The total magnetic field experienced by the nuclear spin would be $B_c\vec{z} + \vec{B}^e$. The net magnetic field experienced by the nuclear spin could be the same as the nuclear spin magnetic field $B^n$ and the only difference is that the direction is opposite. If there is a y disturbing magnetic field $B_y$ input, a total magnetic field $\vec{B}_{tot}$ will be produced and the nuclear spin will also precess to a new direction which is approximately opposite to the total magnetic field direction. The projection of $\vec{B}^n$ in the y direction would be $B_y^n$. We can calculate that $B_y^n$ is equal to $B_y$. This means that the nuclear spins have compensated the disturbing field $B_y$. The electron spins will stay in the z direction and there will be no signal output in the comagnetometer. Note that the electron spins are used to detect the magnetic field produced by the brain or the background magnetic field. However, if there is a brain's magnetic field input whose frequency is very large compared to the comagnetometer's compensation bandwidth, the comagnetometer may not be fast enough to compensate for it. Thus, the comagnetometer could be used for the measurement of the brain's magnetic field as well as being insensitive to the disturbing low-frequency background magnetic field.

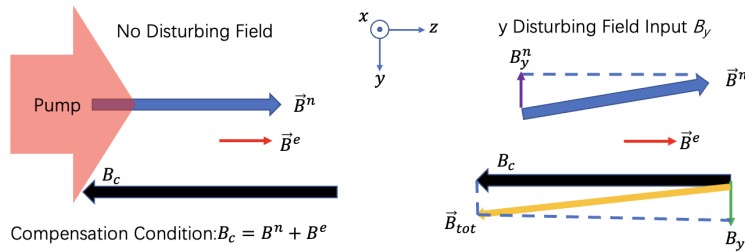

**Figure 3.** The principle of disturbing field compensation by the polarized nuclear spins.

The disturbing magnetic field suppression should also be studied in this paper. The nuclear spin could compensate the disturbing magnetic field. To illustrate this process, we assume that there is sinusoidal disturbing magnetic field input in the comagnetometer $B_y cos(2\pi ft)$ in which $f$ is the frequency of the disturbing magnetic field. The signal output of the comagnetometer $S_y$ is equal to $Ksf_y B_y cos(2\pi ft + \Phi_y)$. Here, $\Phi_y$ is the relative phase between the input and the output signal. $K$ is a factor that links the input magnetic field and the final comagnetometer output voltage signal and it is constant. If there is no suppression process, which means no nuclear spins are used, the signal output could be

$KB_y cos(2\pi ft + \Phi_y)$. The factor $sf_y$ is defined to be the suppression factor of the $B_y$ field which (see [25,40]) is:

$$sf_y = \frac{\omega^2}{[\omega_n^2 + \omega^2 \omega_e^2/(R_{tot}/\gamma^e)^2]},$$ (6)

where $\omega$ is the angular frequency of the input magnetic field and it is equal to $2\pi f$, $\omega_n$ is equal to $\gamma^n B^n$ and $\gamma^n$ is the gyro-magnetic ratio of the nuclear spin, $\omega_e$ is equal to $\gamma^e B^e/Q(P^e)$, $\gamma^e$ and $Q(P^e)$ are the gyro-magnetic ratio of the electron spin and the slow down factor [41], respectively, and $R_{tot}$ is the total relaxation of the electron spins which are shown in the following section. Similarly, we can define the x direction magnetic field suppression factor:

$$sf_x = \frac{\omega}{[\omega_n^2 + \omega^2 \omega_e^2/(R_{tot}/\gamma^e)^2]^{1/2}}.$$ (7)

To further show the compensation process, we show a picture in Figure 4. Here, we chose the typical parameters whereby the temperature of the vapor cell is 420 K. Other parameters such as the nuclear spin polarization and the nuclear spin magnetic field $B^n$ could be determined. We chose the electron spin polarization to be typically 0.5 under which the comagnetometer could utilize the best sensitivity. The magnetic field produced by the electron spins could be:

$$B^e = \frac{8}{3}\pi k_0 \mu_B n_{Rb} P^e,$$ (8)

in which $\mu_B$ is the Bohr Magneton and $n_{Rb}$ is the number density of Rb. The relation between the frequency of the disturbing magnetic field and the suppression factors are shown. As we can see in the frequency range of 1 to 100 Hz, the responses of the comagnetometer to magnetic field is flat and there is about three times the suppression of the magnetic field in the x direction. Moreover, as the frequency changes decrease in size, the suppression factor decreases and the magnetic field is suppressed more efficiently. For the $y$ direction, the magnetic field suppression is larger.

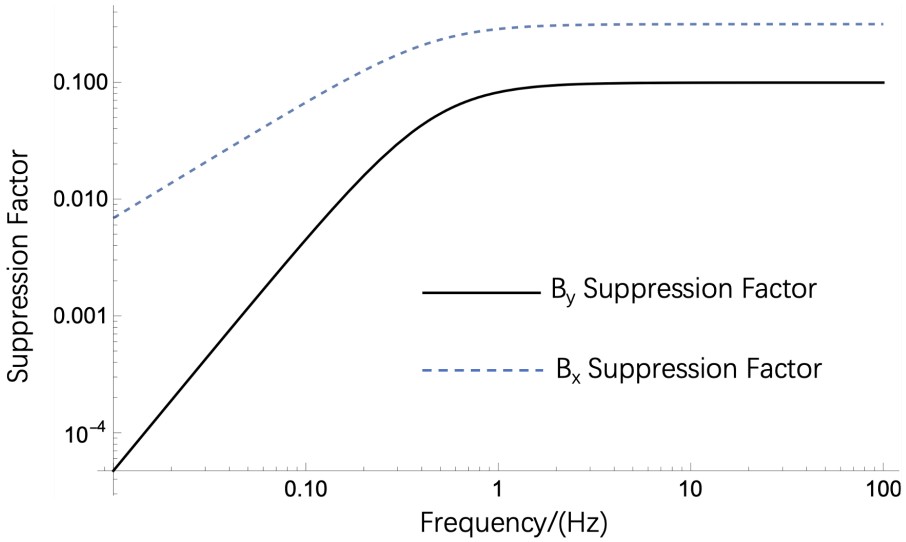

**Figure 4.** Theoretical simulation of the relationship between magnetic field suppression factors of the comagnetometer and the input magnetic disturbing magnetic fields' frequencies.

### 2.3. Relaxation of Rb and Fundamental Sensitivity Estimation

The relaxation of Rb needs to be considered, because it is related to the spin projection noise of comagnetometers. Since Rb is used for the detection of the brain's magnetic field as well as the nuclear spins' compensation magnetic field, the sensitivity of a magnetometer is largely determined by the relaxation of Rb. The relaxation of Rb is caused by several

collision processes, such as collisions with itself, collisions with $N_2$ quenching buffer gas and collisions with $^{83}$Kr nuclear spins. These collision relaxation rates are closely related to the collision cross-sections and number densities of gases. We performed a simulation of the relaxation rates, and Figure 5 shows the results. We changed the temperature of the vapor cell; thus, both Rb–Rb spin destruction relaxation and Rb–$^{83}$Kr spin-exchange relaxation changed with the temperature. The spin destruction rate of Rb–$N_2$ resulted to be weakly temperature-dependent. We also show the relationship between the temperature and the Rb–$^{83}$Kr spin-exchange relaxation rates.

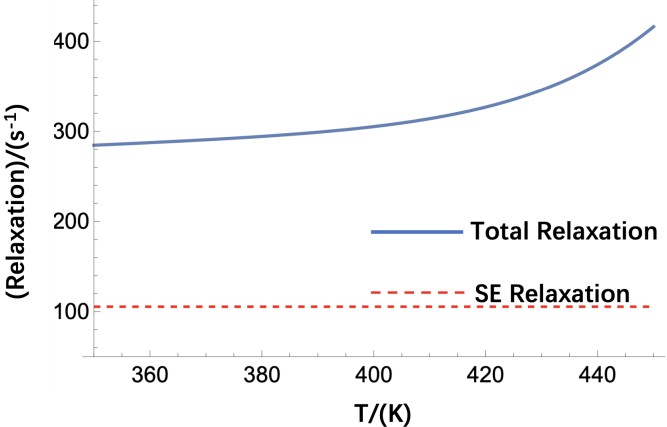

**Figure 5.** Theoretical simulation of the relationship between temperature and the relaxation of Rb, which is used to polarize the nuclear spins. 'SE Relaxation' stands for 'Spin Exchange Relaxation'. Total relaxation includes spin destruction relaxation between Rb and $N_2$, spin destruction relaxation between Rb and Rb, and the spin transfer rate from Rb to $^{83}$Kr. SE relaxation only includes the transfer rate of the Rb electron spin to the nuclear spins.

After we have obtained the total relaxation of Rb electron spins, we can calculate the fundamental sensitivity of a comagnetometer, which is limited by the spin projection noise. The fundamental sensitivity of the comagnetometer is limited by the total relaxation of the atomic spins, the number density of the spins and the volume of the vapor cell. The fundamental sensitivity of a comagnetometer is defined as $\delta B$:

$$\delta B = \frac{1}{\gamma^e} \sqrt{\frac{Q(P^e) 2 R_{tot}}{n_{Rb} V}}, \tag{9}$$

where $\gamma^e$ is the gyro-magnetic ratio of the electron spin. $Q(P^e)$ is the slow-down factor of Rb electron spins [9]. $R_{tot}$ is the total relaxation of Rb spins, which is calculated in the above subsection. Note that there is a factor of 2 here. This factor here means that the optical pumping was also considered. As the optical pumping rate is equal to the total relaxation rate of Rb, the comagnetometer utilizes the best signal-to-noise ratio or sensitivity. Thus, a factor of 2 here considered both optical pumping as well as relaxation's contribution to the spin projection noise. We know that Rb was not directly pumped by the laser. However, we can calculate an equivalent pumping rate which is approximately equal to the optical pumping rate of K divided by the density ratio of Rb to K [42]. $n_{Rb}$ is the number density of Rb, which is related to the temperature of the vapor cell. $V$ is the interaction volume of Rb atoms and laser light. With the equation, we can calculate the fundamental sensitivity of a comagnetometer with changes in temperature and interaction volume. Figure 6 shows the results. We can see that even in a very small volume (0.002 cm$^3$), we can obtain more than 10 fT estimated sensitivity if the temperature of the vapor cell is high enough. This means that we can fabricate very small sensor heads for MEG detection. The space resolution is quite high if we use relatively high temperatures for the vapor cell.

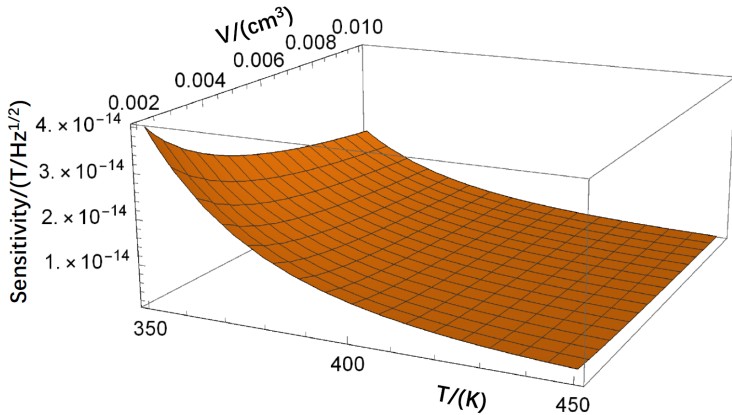

**Figure 6.** Theoretical simulation of the dependent fundamental sensitivity of the comagnetometer on the temperature and the active volume of the vapor cell. The temperature of the vapor cell was changed; thus, the number density of Rb atoms changed. As the temperature increased, we estimated higher sensitivity. We also changed the active volume of the vapor cell in which laser light interacts with Rb atom spins; then, the fundamental estimated sensitivity of the comagnetometer was calculated. As the volume increased, the fundamental sensitivity increased.

Note that the suppression factor could also contribute to the sensor's sensitivity. There is approximately a suppression factor of 3 for the x direction magnetic field in the frequency range of 1–100 Hz. Thus, we believe that the sensitivity of the comagnetometer for detection of the x magnetic field could also be suppressed by a factor of 3. The y direction suppression factor is much larger and thus we think that the comagnetometer is not sensitive to the y magnetic field and it could be treated as a single axis magnetometer. Even though there are some suppression factors for magnetic field detection, we can improve the fundamental sensitivity through increasing the interaction volume of the cell as well as the temperature of the vapor cell.

## 3. Cell Fabrication

### 3.1. K–Rb Mixture Preparation

In order to achieve very high nuclear spin polarization as well as reduce the spin-exchange relaxation of Rb and $^{83}$Kr nuclear spins, hybrid optical pumping is a good solution. If we want to achieve very high nuclear spin polarization, we need to increase the vapor-cell temperature. However, a high-temperature vapor cell could lead to very large optical depth; thus, the laser light might not easily pass through the medium. The polarization of nuclear spins might also be decreased. Hybrid optical pumping is thus needed. The hybrid optical pumping is usually applied in the comagnetometer because its efficiency is higher than a normal optical pump. There are alkali atoms and noble gas atoms in the vapor cell and the electron spin of the alkali metal and the nuclear spin of the inert gas are coupled to achieve hybrid optical pumping. The process is as follows: the electron spin is first pumped by the laser and the nuclear spin is hyperpolarized by spin-exchange collision. Therefore, the polarizability is greatly improved as well as the sensitivity of the comagnetometer. With this technique, we can achieve very high nuclear spin polarization.

We had to prepare the alkali vapor cell before the experiment could be performed. The hybrid optical-pumping cell had to be filled with a mixture of alkali metals. Before the vapor cell was filled with K–Rb mixture using a torch flame on the fabrication apparatus, the K–Rb mixture with the intended Mole Fraction Ratio (MFR) had to be prepared in the glove box. The glove box is an inert gas protection box that can be vacuumed to the main box, so that the system is kept in a high purity inert gas protection environment.

When preparing the K–Rb mixture, the mass ratio of K to Rb had to be determined first. The mass ratio is defined as $M_r = m_K / (m_{Rb} + m_K)$, where $m_K$ is the mass of K and $m_{Rb}$ is the mass of Rb. The mass ratio determines the MFR and the density ratio of K to

Rb in a hybrid pumping cell (HPC) [43]. In a $K - Rb - {}^{87}Kr$ comagnetometer, the density ratio (DR) of K to Rb is controlled for improving the uniformity of spin polarization at the operation temperature. For example, we prepared groups of mixtures in density ratios of 1/212 and 1/109 at 463 K after the cell was filled with them. In order to reach the intended DR, Table 1 shows the masses of K and Rb that had to be mixed together in the glove box.

**Table 1.** The calculation of the masses of K and Rb that had to be mixed together to reach the intended $D_r$.

| Intended $D_r$@463 K | $M_r$ | $m_K$ (mg) | $m_{Rb}$ (g) | MFR of K |
|---|---|---|---|---|
| 1/212 | 0.014 | 30 | 2.13 | 0.029 |
| 1/109 | 0.026 | 41 | 1.54 | 0.055 |

The calculation of the mass of K and Rb determines the amount of alkali metal that should be mixed together in the glove box. Here, before the mixing of alkali metals, the glassware used in this experiment was cleaned before being put into the glove box to protect the alkali metals against impurities. All of the glassware was immersed into piranha solution and cleaned with an ultrasonic cleaner for 5 minutes. Several rounds of rinsing with deionized water were employed to rinse away the residual piranha solution. After cleaning it, we quickly put the glassware into a drying box and left it there for several hours to remove any water on the glassware. When the glassware was put into the transfer chamber of the glove box, the gases in the chamber were pumped away to further dry the glassware.

The mixing of K–Rb metals was performed after a low concentration of oxygen was achieved in the glove box. Commercial Rb and K break-seal ampoules were employed. A homemade glass spoon was used to scoop Rb and K metals out of the ampoules. With a glass knife, we cut the metals into small pieces in a Petri dish. A piece of weighting paper was put on the analytical balance to weigh the pieces of alkali metals. According the calculation results in Table 1, 30 mg of K and 2.13 g of Rb were added to a homemade break-seal ampoule to obtain 1/212 $D_r$ at 463 K in a hybrid pumping cell. A Teflon valve and an O-ring sealed the alkali metal mixture in the ampoule to protect the mixture against air when the ampoule was taken out of the glove box. To seal the mixture in the glass ampoule and protect the mixture against air, a vacuum pump was used to pump out the gases in the ampoule. Methane flame was used to seal the glass at the neck of the ampoule during pumping. The mixture was finally sealed in the glass ampoule. Figure 7 shows this process. For HPCs with small MFRs, we used a glass knife to cut the alkali metal into small pieces of approximately 30 mg. Moreover, we used a valve-and-pumping technique to protect the mixture against oxygen and water after the ampoules were moved out of the glass box.

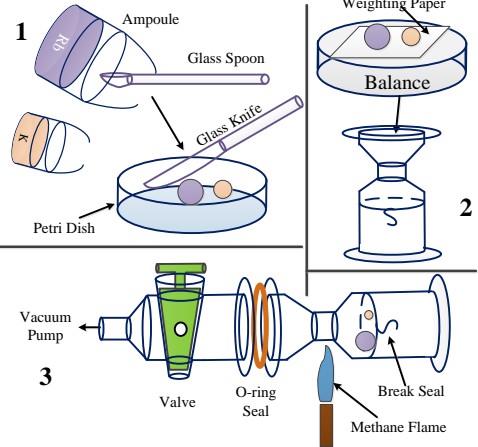

**Figure 7.** The process of experimentally preparing the K–Rb mixture in a glove box: 1. The cutting of the alkali metals. 2. The weighting of the alkali metals. 3. The sealing of the mixed alkali metals.

### 3.2. Alkali Chasing and Gas Filling

The mixture break-seal ampoule was attached to the alkali vapor-cell fabrication apparatus, and the cell fabrication process began when the ampoule was ready. Figure 8 shows the method of filling the vapor cells with the K–Rb mixture. The K–Rb mixture ampoule was connected to the alkali fabrication apparatus with a glass tube. An iron hammer, which was connected with the ampoule to break the seal, was sealed in the vertical glass tube. The vapor cell was connected to the glass tube, and the glass tube was connected to the vacuum pumping system and the gas line system. The vacuum pumping system pumped the whole system, and the gas line system supplied the vapor cell with gases such as $^{21}Ne$, $^{131}Xe$, $^{129}Xe$, $^{83}Kr$, $N_2$. The gas line system and the vacuum pumping system were similar to the system described in reference [44]. All the glassware was cleaned with piranha solution and deionized water, like the glassware used to make the mixture ampoule. To further clean the surface of the glassware, after it was connected to the fabrication system, an oven was used to heat the glassware to about 473 K when the pumping system was on. After 2 to 3 days of heating and pumping, the vacuum pressure reached the $10^{-5}$ Pa level. The vapor cells were filled with K–Rb mixture when the vacuum reached the $10^{-5}$ Pa level. Then, the K–Rb mixture was slowly chased into the retort using a flame for thoroughly mixing the K–Rb alloy and purifying the alloy. The ampoule fired off at the small neck of the glass tube after the mixture was chased into the retort. After that, the mixture was chased into the vapor cell, and the experimental results show that Rb moved faster than K due to the lower melting point of Rb. As a result, a special chasing method was developed to ensure that the MFR in the HPC was right. Most of the mixture was chased into the cell and then chased out of the cell to leave the amount of alloy that was needed. This process ensured that the MFR in the cell was close to the MFR in the ampoule. Figure 8 shows the schematic of the filling process.

After the vapor cells were filled with alkali metal mixtures, the vapor cells were filled with gases through the gas line. In some of the experiments, the vapor cell was filled with gases at about 2 Atm. For sealing the glass using a flame, the cell cannot be sealed until the inner pressure is lower than the pressure outside. Liquid nitrogen was used to fill the cell with gases at more than 1 Atm.

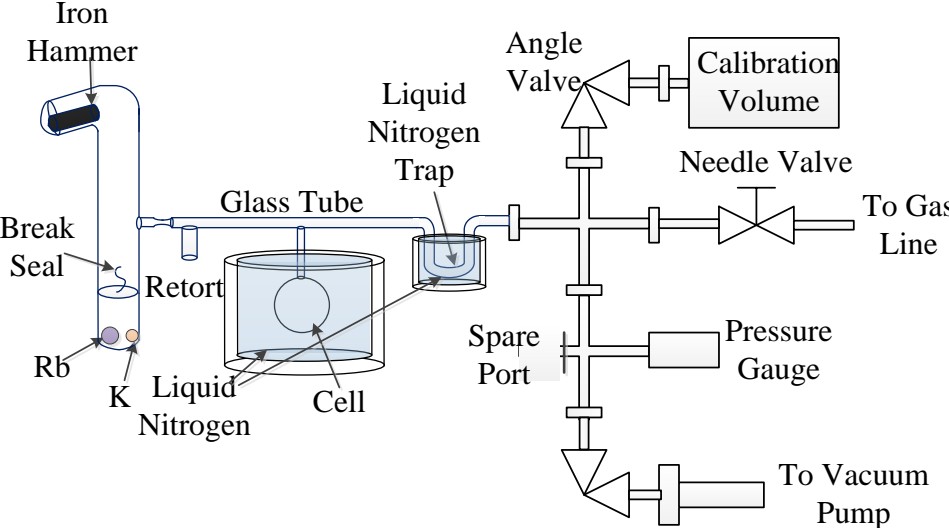

**Figure 8.** Schematic of filling the vapor cell with K–Rb mixture and gases experimentally.

Figure 8 shows the schematic of filling the glass cell with gas with a pressure more than 1 Atm. The liquid nitrogen trap was taken down, and the temperature of the trap rose to room temperature before gas filling. The needle valve allowed the gas to flow into the cell, and the pressure gauge measured the pressure in the cell. When the intended pressure ($P_0$) was reached, the gas line was shut down. A homemade Dewar was filled

with liquid nitrogen, and the Dewar was lifted with a lifting platform to immerse the cell into liquid nitrogen. The pressure decreased and settled down to a lower pressure ($P_1$). A torch flame was used to fire off the stem of the cell, and the pressure rose slightly to $P_2$. This phenomenon was due to the increase in the temperature of the cell when the flame was used to fire off the cell from the glass tube. In order to know exactly the pressure of the cell after it was fired off from the glass tube and the temperature of the cell increased to room temperature, the ideal gas equation was used to calculate the pressure. The volume of the cell was measured to be $V_c$, and the volume of the glass tube and the gas line tube was measured to be $V_g$ (the measurement method is described below). The pressure of the cell at room temperature should be

$$P_c = P_0 + (P_0 - P_2)\frac{V_g}{V_c} \tag{10}$$

Further, $V_g$ was calculated with the ideal gas equation, and standard calibration volume was used for the calculation. We supposed that the standard calibration volume was $V_{cal}$ and that the calibration volume was pumped to vacuum state. $V_g$ was filled with gas, and the pressure was $P_{g0}$. After opening the angle valve of the calibration volume, the pressure changed to $P_{g1}$. The volume of the glass tube and the gas line was calculated to be

$$V_g = \frac{P_{g1}V_{cal}}{P_{g0} - P_{g1}} \tag{11}$$

After volume $V_g$ was measured, cell pressure $P_c$ was finally calculated with Equation (10). $P_c$ is the pressure of the gases in the vapor cell at room temperature.

## 4. Discussion

This paper theoretically studies a $^{83}$Kr-based atomic comagnetometer, which is a promising device for movement-related MEG recording. The main applications of moving MEG include the diagnosis of neuropsychiatric diseases such as epilepsy and Alzheimer's disease, the determination of brain function injury, and the localization of brain function before craniocerebral surgery. Moving MEG can also be used to explore brain functions, such as what people are thinking while drinking coffee or playing the guitar.

Due to the large Fermi contact interaction with Rb electron spins, the compensation field produced by the nuclear spin is quite large. A wide compensation range can be achieved, which is of utmost importance for effectively compensating for fluctuations in background magnetic fields in the future. Due to the small destruction collision relaxation rate of $^{83}$Kr atoms, Rb magnetometers could achieve very high sensitivity. An experimental setup is under way to configure a K–Rb–$^{83}$Kr comagnetometer.

The temperature of the vapor cell designed in this study is 420 K and around 760 Torr $N_2$ is filled in the vapor cell. The condition here is quite different from the typical $^{83}$Kr polarization experiment in which the temperature of the vapor cell is around 373 K and only several tens of Torr $N_2$ is filled in the vapor cell. The polarization of $^{83}$Kr is typically smaller than 10% [33,45]. With the high density of Rb, the polarization of $^{83}$Kr could nearly reach that of the Rb polarization. The high pressure of $N_2$ in our design also eliminates the relaxation of Rb-$^{83}$Kr due to the formation of van de Waals molecules. The characteristic spin-relaxation rate due to the van der Waals molecule $\gamma_M$ is only 63 sec$^{-1}$ [33] which is much smaller than that of the Rb-Xe pair [46]. That is because Rb-Xe pair could form van der Waals molecules more easily. $N_2$ could also affect the formation of the molecule and we fill the high pressure of $N_2$ to stop the formation of the molecules in our design. According to this reference [46], the spin exchange rate coefficient due to van der Waals molecule $\kappa_T$ is $(\gamma_M \zeta)/([Kr](1+br))$. $\zeta = 0.179$ is related to the ratio between the two isotopes of Rb. $[Kr]$ is the number density of Kr. $r = P(N_2)/P(Kr)$ is the ratio of the $N_2$ pressure $P(N_2)$ to the Kr pressure $P(Kr)$ which is around 15 in our design. The coefficient $b = P_0(N_2)/P_0(Kr) = 1.9$, where $P_0(N_2)$ and $P_0(Kr)$ are the characteristic pressures defined in the reference [33]. We

finally could calculate that $\kappa_T$ is an order smaller than that of the binary spin exchange rate coefficient $\sigma_{se}\bar{v}$. Thus, we could neglect the van der Waals molecule formation-related spin exchange optical pumping.

We know that comagnetometers are also sensitive to rotation velocity. For the $^{83}$Kr-based atomic comagnetometer, the nuclear spin is 9/2, and we estimated a very small gyro-magnetic ratio, which means that this comagnetometer is quite sensitive to rotation. When performing movement-related MEG, we can only measure movements without rotation. Further experiments need to be performed to study this topic. The study of $^{83}$Kr-based atomic comagnetometers also contributes to the study of gyroscopes.

Relaxation rate, polarization and fundamental sensitivity are important parameters for atomic comagnetometers. Note that $^{83}$Kr nuclear spin relaxation is quite complicated, since quadrupolar relaxation at the wall as well as the nuclear spin itself should be considered. Especially, collisions with the wall, desorption energy and collision-related average angle should be studied. The quadrupolar interaction process makes simulations more complicated. Our results give very concrete evidence for future comagnetometer design.

The sensitivity of the comagnetometer is fundamentally determined by Equation (9). It is mainly determined by the total relaxation of Rb $R_{tot}$ and the number of the Rb $n_{Rb}V$ which are used for the magnetic field detection. As we know, we want $n_{Rb}V$ to be as small as possible and thus we can fabricate very small alkali vapor cells. The size of the sensor could be smaller and the spatial resolution of magnetic field measurement could be improved. The total relaxation of Rb $R_{tot}$ is determined by several collision relaxation processes. The merit of using $^{83}$Kr is that the spin collision relaxation with $^{83}$Kr is quite small compared with that of the Rb-$^{129}$Xe collision. To further show this clearly, we have conducted a simulation shown in Figure 6. The results show that even with very small volume (0.002 cm$^3$), the estimated sensitivity of the comagnetometer could reach higher than 10 fT/Hz$^{1/2}$.

In the last part of the paper, we show the alkali fabrication process. It is especially noteworthy that hybrid optical pumping is needed. We give details on how to make a hybrid-pumping vapor cell. We also present the alkali fabrication process with the glass-blowing technique.

With the development of the micro-electro mechanical system (MEMS), the key elements of the comagnetometer such as alkali vapor cell, micro-electrical non-magnetic field heating chips, optical lens and the semiconductor laser diodes can be fabricated into chip-scale. Therefore, the size of the future comagnetometer will be smaller and smaller. As we have calculated, we only need 0.002 cm$^3$ effective alkali vapor cell space for the comagnetometer. Other elements of the comagnetometer which need to be designed around the vapor cell could also be fabricated to be very small. The comagnetoemter could finally be fabricated in chip-scale.

In comparison, the $^{83}$Kr-based atomic comagnetometer could work under 420 K and the compensation bandwidth is around 1 Hz which is similar to that of the $^{21}Ne$ comagnetometer [47]. However, the $^{21}Ne$-based atomic comagnetometer needs to work under around 473 K. Special aluminosilicate glass is required for the cell fabrication. This makes the experiment complicated since it is not easy to obtain the aluminosilicate glass. There are around 3 Amagats $^{21}Ne$ filled in the vapor cell, while there is only 50 Torr $^{83}$Kr filled in the vapor cell. Again, it is easier to fabricate a vapor cell based on $^{83}$Kr. The total relaxation of Rb which could determine the sensitivity of the comagnetometer is around 300 s$^{-1}$ in the $^{83}$Kr atomic comagnetometer which is only around 1/6 of that of the $^{21}Ne$ comagnetometer [47]. This would make sure that the sensitivity of the $^{83}$Kr-based atomic comagnetometer will be higher.

## 5. Conclusions

In conclusion, we have theoretically designed a new kind of atomic comagnetometer that is based on $^{83}$Kr nuclear spins. Compared with $^{21}$Ne-, $^{3}$He- and $^{129}$Xe-based atomic comagnetometers, the new comagnetometer has the merits of high sensitivity, relatively low temperature and a large compensation field range. We conclude that the $^{83}$Kr-based

comagnetometer is quite promising for movement-related MEG recording. We give the key parameters for the design of the comagnetometer, such as spin relaxation and polarization. These parameters are highly important for future experiments. Considering that the alkali vapor cell is important for the comagnetometer, we show a method for developing hybrid optical-pumping vapor cells. The study presented in this paper is highly valuable for a future experimental setup.

**Author Contributions:** Conceptualization, Y.C.; methodology, R.G. and J.W.; investigation, M.Y. and M.Z.; writing—original draft preparation, Y.C.; supervision, L.Z. All authors have read and agreed to the published version of the manuscript.

**Funding:** This work was supported by National Natural Science Foundation of China under grant number 62103324, China Postdoctoral Science Foundation under grant number 2020M683462, National key research and development program under grant number 2022YFB3203400.

**Institutional Review Board Statement:** Not applicable.

**Informed Consent Statement:** Not applicable.

**Data Availability Statement:** The data underlying the results presented in this paper are not publicly available at this time but may be obtained from the authors upon reasonable request.

**Acknowledgments:** We want to thank all of these agencies mentioned above for their support.

**Conflicts of Interest:** The authors declare no conflict of interest.

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
