# Peer review of "Theoretical Study on Performing Movement-Related MEG with 83Kr-Based Atomic Comagnetometer"

_photonics, doi:10.3390/photonics10121302_

Round 1

Reviewer 1 Report

This manuscript has developed a method for moving MEGs recording which is very promising for the future brain’s magnetic field measurement. It is very good for moving related brain function study. At the same time, the authors have established a 83Kr based atomic co-magnetometer. Traditionally, 21Ne, 3He based atomic co-magnetometers have been developed. Atomic co-magnetometers based on new isotope enriched nuclear spins are quite important for the field of atomic co-magnetometer and gyroscope as well as for brain magnetic field sensing. It is quite meaningful to study this problem. In this manuscript, the authors have shown the quadrupolar relaxation of 83Kr based atomic co-magnetometer. It is very important to show this and it is very hard to study the quadrupolar interaction because the interaction is pretty complicated.

  In my opinion, this paper has shown a very meaningful and important technique for the future application. I only have few comments before the paper could be accepted.

Comments 1: At line 70, the structure of the paper should be shown.

Comments 2: In figure 4, I notice that the volume of the simulation is quite small. In traditional vapor cell, the volume is around 1cm3, however, the simulation here is quite small volume. What is the reason?

Comments 3: In the discussion section, the authors should show the fundamental sensitivity of the co-magnetometer.

Minor editing of English language required.

Reviewer 3 Report

Review of ‘Theoretical study of measuring moving MEGs with 83Kr based atomic co-magnetometer’ by Chen et al.

The authors present a theoretical overview of a co-magnetometer with a high magnetic field compensation rate, within the context of performing magnetoencephalography measurements. Whilst an interesting concept, the paper would benefit from expansion on key points.

I appreciate the reasoning behind the heavy emphasis on MEG in the paper, but I feel the background and aims need to be tightened to ensure a MEG researcher reading the paper can follow its findings. For example, it is not entirely clear how the nucelar spin generates magnetic fields for compensation of field changes within the cell. Other simple things like, is the minima of the curve shown in Figure 1 a good or bad place to operate the sensor? A schematic diagram of concept of using nuclear spins to compensate fields? How the theoretical sensor would operate including pump and probe laser directions and polariations? What is the bandwidth of the compensation in the context of MEG and how does this vary with sensitivity and other trade offs?

The content is similar to the paper ‘Spin exchange optically pumped nuclear spin self compensation system for moving magnetoencephalography measurement’ by many of the same authors (Biomedial Optics Express Vol. 13, No. 11, 2022). Whilst this other study used Neon in place of Krypton, the authors did construct a working magnetometer and evaluate the suppression. In light of this, perhaps adding brief comparison to a Neon system and what the performance increase of a Kr sensor is would help to highlight the novelty of what is presented here. Even better would be a working magnetometer to properly tie together the two sections of this paper which at present feels like a study which is only partly complete without significant changes.

Other comments:

Reference [15] is tackling a different problem which is head movement relative to a fixed array of OPMs, as opposed to the OPM issue of a moving array of sensors which is fixed to the head.

Acronym MEGs should be defined in the text.

It is not just the geomagnetic field that effects MEG measurements. Also implies EEGs are not recorded on Earth…

For the fabrication, define a glove box?

Some phrases including ‘technique problem’ ‘gives’ ‘owns’ should be revised. 

Reviewer 4 Report

Major

-          The paper sets up the problem with background magnetic fields for MEG quite well. Indeed, this is a limiting factor, and several groups are working to supress this with active coil systems, as the authors cite. However, the paper does a poor job at explaining why the addition of a hyperpolarised nuclear spin can create a compensation effect. The authors cite their previous paper on this topic, but this is insufficient for the reader to then understand the rest of the paper. I strongly advise the authors to summarise the motivation clearly in the introduction.

-        There is a certain effort at the end of the introduction to distinguish 87Kr as the isotope of choice over Ne etc, but I found it rather diffuse. Some numerical data would be useful here, e.g roughly how much more expensive is 21Ne? How much more polarisable is 87Kr than 3He? A clearer explanation of the benefit of Kr over the other options would increase the impact of this paper.

Minor

co-magnetometer: I believe the convention is to say 'comagnetometer'

MEGs: why is there an s? I think the convention is just 'MEG' as this is not plural

-          Line 20: OPMs*

-          Define MEG in the abstract

-          Line 45: complicated*

-          Line 78: quadrupolar*

-          Line 88: an order (of magnitude) higher?

-          Numerical values such as ‘50Torr’ should have a space e.g 50 torr. This error is repeated further on in the text.

-          Equation 6: should this not be *delta* B = … ?

-          Line 167: volume*

-          Make the acronym for density ratio = DR rather than Dr which is a title.

Generally ok, some plurals mistaken for singulars and vice versa. 

Reviewer 5 Report

I noticed a few minor typos. The authors may consider the following suggestions

line 50 "could compensation" -> "could compensate"

line 51 "automatically compensated" -> "automatically compensate"

line 81 "the faster of the relaxation" -> "the faster is the relaxation"

line 139 "We can estimate that the larger the nuclear spin magnetic field, the higher compensation ability of the nuclear spins." -> "We can estimate that the larger the nuclear spin magnetic field is, the higher the compensation ability of the nuclear spins is."